# Biomechanical Properties of Strictures in Crohn’s Disease: Can Dynamic Contrast-Enhanced Ultrasonography and Magnetic Resonance Enterography Predict Stiffness?

**DOI:** 10.3390/diagnostics12061370

**Published:** 2022-06-02

**Authors:** Rune Wilkens, Dong-Hua Liao, Hans Gregersen, Henning Glerup, David A. Peters, Charlotte Buchard, Anders Tøttrup, Klaus Krogh

**Affiliations:** 1Diagnostic Centre, Divisions of Medicine and Radiology, University Research Clinic for Innovative Patient Pathways, Silkeborg Regional Hospital, Falkevej 1-3, 8600 Silkeborg, Denmark; hengle@rm.dk; 2Department of Hepatology and Gastroenterology, Aarhus University Hospital, Palle Juul-Jensens Blvd. 99, 8200 Aarhus N, Denmark; klaukrog@rm.dk; 3Digestive Disease Center, Copenhagen University Hospital—Bispebjerg and Frederiksberg, Bispebjerg Bakke 23, 2400 Copenhagen NV, Denmark; 4Mech-Sense, Department of Gastroenterology and Hepatology, Aalborg University Hospital, Hobrovej 18-22, 9000 Aalborg, Denmark; dl@rn.dk; 5GIOME, California Medical Innovations Institute, 11107 Roselle St., San Diego, CA 92121, USA; hag@giome.org; 6Department of Clinical Engineering, Central Denmark Region, Universitetsbyen 25, Bygning 2A, 3. Sal, 8000 Aarhus C, Denmark; david.peters@stab.rm.dk; 7Department of Surgery, Aarhus University Hospital, Aarhus University Hospital, Palle Juul-Jensens Blvd. 99, 8200 Aarhus N, Denmark; buchard@dadlnett.dk (C.B.); andetoet@rm.dk (A.T.)

**Keywords:** Crohn’s disease, stricture, stiffness, medical imaging, ultrasound, MRI, fibrosis

## Abstract

Strictures and abdominal pain often complicate Crohn’s disease (CD). The primary aim was to explore whether parameters obtained by preoperative contrast-enhanced (CE) ultrasonography (US) and dynamic CE MR Enterography (DCE-MRE) of strictures associates with biomechanical properties. CD patients undergoing elective small intestinal surgery were preoperatively examined with DCE-MRE and CEUS. The excised intestine was distended utilizing a pressure bag. Luminal and outer bowel wall cross-sectional areas were measured with US. The circumferential stricture stiffness (Young’s modulus *E*) was computed. Stiffness was associated with the initial slope of enhancement on DCE-MRE (*ρ* = 0.63, *p* = 0.007), reflecting active disease, but lacked association with CEUS parameters. For structural imaging parameters, inflammation and stricture stiffness were associated with prestenotic dilatation on US (τ_b_ = 0.43, *p* = 0.02) but not with MRE (τ_b_ = 0.01, *p* = 1.0). Strictures identified by US were stiffer, 16.8 (14.0–20.1) kPa, than those graded as no or uncertain strictures, 12.6 (10.5–15.1) kPa, *p* = 0.02. MRE global score (activity) was associated with *E* (*ρ* = 0.55, *p* = 0.018). Elastography did not correlate with circumferential stiffness. We conclude that increasing activity defined by the initial slope of enhancement on DCE-MRE and MRE global score were associated with stricture stiffness. Prestenotic dilatation on US could be a potential biomarker of CD small intestinal stricture stiffness.

## 1. Introduction

The natural course of Crohn’s Disease (CD) usually evolves from active inflammation to stricturing and penetrating disease [1,2]. Active inflammation is mainly treated with medical therapy aiming at mucosal healing and normalization of bowel habits without pain [3,4]. Unfortunately, there is still a lack of evidence for successful medical treatment of fibrotic strictures [5,6]. Consequently, fibrotic strictures and penetrating disease are considered the main indications for small intestinal surgery in CD [7]. Persistent inefficient medical treatment is expensive and may lead to corticosteroid dependency or abdominal pain [8]. 

Strictures are defined from cross-sectional imaging and endoscopy as localized, persistent lumen narrowing with or without prestenotic dilatation [9,10,11]. Strictures pose a mechanical challenge for normal intestinal transport function since it dramatically increases the resistance to flow. The resultant proximal dilation induces impaired intestinal function and pain. Hence, quantifying intestinal strictures’ mechanical properties is important [12,13,14,15]. Cross-sectional imaging modalities relatively easily detect small intestinal strictures in CD [16,17]. A commonly used definition is wall thickening and prestenotic dilatation >25 mm after oral contrast combined with luminal narrowing <10 mm [11,18]. Clinically, we often classify strictures as symptomatic or asymptomatic. Unfortunately, such a classification is highly subjective and depends on several factors, including food composition [19,20]. Furthermore, no consensus exists on stricture classification in clinical trials [11,17,21,22,23]. Hence, objective methods for evaluating the properties of strictures in CD are warranted.

Conventional imaging inadequately separates strictures with active inflammation, mixed inflammation with fibrosis, and fibrosis without significant inflammation [11,17,24]. Magnetic Resonance Enterography (MRE) and Ultrasonography (US) have high sensitivity and specificity for detecting strictures based on structural findings [16,17]. With contrast-enhanced US (CEUS), higher contrast enhancement is found in segments with inflammation than in those with fibrosis [25,26]. Furthermore, the percentage of enhancement gain on MRE after seven minutes is associated with the degree of fibrosis [27]. The abovementioned studies suggest that dynamic contrast enhancement helps differentiate inflammatory and fibrotic strictures. However, we remain to understand how CEUS and Dynamic Contrast-Enhanced MRE (DCE-MRE) are associated with the biomechanical properties of small intestinal strictures.

Mechanical properties of the gastrointestinal tract are essential for both normal and abnormal organ function, and well-established accurate and reproducible methods for assessment in vitro exist [12,13,14,15]. Abnormal mechanical properties may cause symptoms, and some diseases change the mechanical properties. The extracellular matrix stiffness is an independent activator of fibroblasts [28], and the longitudinal tensiometry strain (stretch) of strictures is inversely associated with US elastography and fibrosis [29]. Tsamis et al. computed the theoretical stress and strain in strictureplasty surgery [30]. Despite this, our knowledge about the mechanical properties of small intestinal strictures and assessing the stiffness in vivo is minimal.

The present study aimed to explore which parameters obtained by preoperatively CEUS and DCE-MRE of small bowel CD are associated with the biomechanical properties of the strictures. Our primary hypothesis was that strictures with low perfusion on CEUS and DCE-MRE are stiffer than those with high perfusion, as the former would be fibrotic and the latter inflammatory [25,26].

## 2. Materials and Methods

The present study was a GCP-monitored blinded prospective observational study. Patients were 18 years or older and had small intestinal inflammation or stricture on US, as described in a previous publication [31]. A flowchart of patients included is shown in Figure 1, and demographics in Appendix A. Data on associations between imaging methods, perfusion, stiffness, and histology from the same group of patients have been published previously [31,32].

### 2.1. Cross-Sectional Imaging

The preoperative imaging procedures have previously been described in detail [31,33]. In brief, intestinal US was performed using an Acuson S3000™ with a 9L4 linear matrix probe (Siemens Medical Solutions, Malvern, PA, USA) after the patient had fasted for at least four hours. CEUS was continuously recorded for 90 s after a dual injection of 2.4 mL sulphur hexafluoride (SonoVue^®^; Bracco Imaging, Milan, Italy). Time-intensity curves were analyzed in VueBox^®^ 5.1 (Bracco Suisse SA, Geneva, Switzerland). The mean values of three ROIs from each of the two injections were used to compare biomechanical parameters since this has been shown to achieve the best repeatability [33]. In addition, we performed point shear-wave elastography with the built-in acoustic radiation force impulse utilizing the same 9L4 probe. The region of interest was 6 × 5 mm and placed inside the bowel wall. The median value of 10 measurements was used for further analysis.

MRE is performed routinely in our department using 4 h fast and ingestion of 1 L Mannitol solution. Patients lie in the prone position throughout the scan. 20 mg of Hyoscine butyl bromide (Buscopan^®^; Boehringer Ingelheim, Ingelheim, Germany) were injected intravenously before scanning non-dynamic sequences. This was repeated immediately before dynamic contrast sequences. In addition to the DCE-MRE analysis, components of the MR enterography global score (MEGS) [34] and MR index of activity (MaRIA) [35] were included, and the total scores calculated.

All patients included in the study had bowel wall thickness >3 mm. No clear definition exists for strictures [11,22,23]. For both imaging modalities, stenotic area was defined as lumen <10 mm, and prestenotic dilatation was graded as none (lumen of inflamed area >10 mm); mild (prestenotic diameter greater than stricture but no more than adjacent loops), moderate (prestenotic diameter greater than adjacent loops), severe (prestenotic diameter >25 mm). Strictures were regarded as “certain” for moderate and severe prestenotic dilatation in both imaging modalities and “no stricture/uncertain” for the remaining cases. At inclusion, patients were assessed with the Harvey-Bradshaw index, and their symptom severity was graded into the four categories: no, mild, moderate, and severe symptoms.

### 2.2. Distension and Ex Vivo Ultrasonography

Immediately after surgical resection, the bowel specimen was opened at the stapled ends, rinsed in lukewarm water, and placed in a water tank containing 37 °C buffered Krebs Solution with 100% O_2_ 1 L/min. The Krebs solution included 118 mmol/L NaCl, 4.7 mmol/L KCl, 25 mmol/L NaHCO_3_, 1.0 mmol/L NaH_2_PO_4_, 1.2 mmol/L MgSO_4_, 2.5 mmol/L CaCl_2_-H_2_O, 11 mmol/L glucose, and 0.11 mmol/L ascorbic acid [36]. The water tank was heated using an electric warming tray (Bartscher, Salzkotten, Germany). A 3 mm catheter with a 12 cm long bag mounted towards the tip (Endoflip^®^, Crospon, Galway, Ireland) was inserted from the anal side of the bowel segment. The bag was slightly distended, and we utilized a HI VISION Preirus US machine with a EUP-L73S probe (Hitachi Medical Corporation, Tokyo, Japan) to obtain b-mode images to determine the most stenotic part of the specimen by measuring the lumen diameter [15]. The bowel and the inserted catheter were fixated at both ends after deflation and relocation to ensure optimal positioning (Figure 2). A tiny acupuncture needle was inserted in the cross-sectional direction in the mesentery or serosa at the most stenotic area. The needle, easily identified on ultrasound, ensured the same location of consecutive ultrasound images. The catheter was connected to a fluid column for pressurizing the segment. The level container was aligned with the catheter to obtain zero pressure. B-mode ultrasound images at each site of needle placement were stored and annotated with 0, +1, +2, or −1, −2 cm and pressure level, Appendix A. After the five measurements were obtained, the water column was elevated 10 cm, and the next images were obtained after pressure equilibrium was obtained. We repeated the procedure with 10 cm increments until 100 cmH_2_O pressure.

### 2.3. Data Analysis

We only analyzed the narrowest and stiffest part of the inflammation/stricture since this is the primary determinant of flow through a stricture [12]. US images were analyzed using OsiriX 5.7.1. 64-bit (Pixmeo SARL, Bernex, Switzerland). A circular/elliptical region of interest (ROI) was drawn for the luminal and muscularis propria bowel wall border. ROIs were named according to the picture annotations, and ROI data were exported in .csv files and imported into MATLAB^®^ (R2017b, MathWorks^®^, Natick, MA, USA) for further analysis.

The bowel segments were assumed to be straight cylinders. The ultrasound images were analyzed for circumference (*C*) and luminal areas (*LA*) of inner and outer surfaces. At each location, the circumferential Green strain (*ε_θ_*) and circumferential Kirchhoff stress (*S_θ_*) for each distension pressure were calculated as [13,14,15,37]:

Circumferential Green strain:(1)$$\varepsilon_{\theta }^{}=\frac{1}{2}(\lambda_{\theta }^{}{}^{2}-1)$$

Circumferential Kirchhoff stress:(2)$$S_{\theta }^{}=\frac{\Delta P\cdot r_{inner}^{}}{\lambda_{\theta }^{}{}^{2}\cdot h^{}}$$
where $\lambda_{\theta }^{}=\frac{\left(C_{inner}^{}+C_{outer}^{}\right)}{\left(C_{inner}^{0}+C_{outer}^{0}\right)}$ is circumferential stretch ratio, and *C_inner_*, *C_outer_*, are circumference for inner and outer surface in the pressurized state, $C_{inner}^{0},C_{outer}^{0}$ are circumference at pressure 0 cmH_2_O. ΔP is the transmural pressure difference, $r_{inner}^{}=\sqrt{LA_{inner}^{}/\pi }$ is the inner radius calculated from the measured luminal area of the inner surface (*LA_inner_*), $r_{outer}^{}=\sqrt{LA_{outer}^{}/\pi }$ is the outer radius calculated from the outer surface (*LA_outer_*), and $h^{}=r_{outer}^{}-r_{inner}^{}$ is wall thickness. The calculated stress-strain curves were fitted to the exponential function:(3)$$S_{\theta }^{}=\beta (e^{\alpha \varepsilon_{\theta }}-1)$$
where *α* and *β* are material constants, and α relates to the wall stiffness. The circumferential Young’s modulus (*E*) was thus calculated as:(4)$$E=\frac{dS_{\theta }}{d\varepsilon_{\theta }}$$

In this paper, *E* for 2.5 < *S_θ_* < 5 kPa was calculated, assuming linearity.

### 2.4. Statistical Analysis

Descriptive statistics were used for all demographic variables. These included means, standard deviations, and ranges for continuous variables and frequencies and percentages for categorical factors. Several parameters did not follow the Gaussian distribution. Young’s Modulus was log-converted for further analysis, and CEUS intensity parameters (expressed as arbitrary intensity units (AIU)) were converted to dB using the equation 10 × log_10_ (AIU) = dB. Spearman’s correlation (*ρ*) was computed between *E* and CEUS or DCE-MRE parameters. Furthermore, spearman’s correlation was used for associations between *E* and MEGS, MaRIA. For ordinal categorical variables, including the degree of prestenotic dilatation, MRE mural edema, and enhancement pattern, Kendall’s τ_b_ was computed. Categorical variables were examined with one-way ANOVA before the investigation of pairwise differences. Finally, the difference in *E* was tested with Students t-test for dichotomous variables such as the presence of strictures and ulcers. A *p*-value < 0.05 was regarded as significant.

## 3. Results

Among 25 patients included [31], distensibility data were available for 18 (Figure 1, Appendix A). 

### 3.1. Stiffness of the Inflammation/Stricture and Dynamic Contrast-Enhanced MR Enterography or Contrast-Enhanced Ultrasonography

Averaged circumferential stress-strain curves are shown in Figure 3. Whether classified as “certain” strictures with MRE (*n* = 11) or US (*n* = 10), the stress-strain curves from stenotic segments were located to the left of those from non-stricture segments, illustrating the stiffer wall of the strictures. A strong association was found between *E* and DCE-MRE Initial slope of enhancement (*ρ* = 0.63, *p* = 0.007, Figure 4). No significant correlation was found between *E* and any of the CEUS parameters. All correlation parameters are listed in Appendix A. 

### 3.2. Stiffness of Inflammation/Strictures and In Vivo Ultrasonography or MR Enterography

Young’s modulus for the stricture was moderately associated with the grading of prestenotic dilatation (none; mild; moderate; severe) when assessed before surgery by US (τ_b_ = 0.43, *p* = 0.02) but not by MRE (τ_b_ = 0.01, *p* = 1.0, Figure 5). Strictures classified as “certain” on in vivo US were stiffer, *E*: 16.8 (14.0–20.1) kPa, than those classified as “no” or “uncertain,” *E*: 12.6 (10.5–15.1) kPa, *p* = 0.022. No such difference was found between the certainty of strictures on MRE and *E*: 15.4 (12.5–19.0) kPa for “certain” strictures and 13.8 (11.4–16.7) kPa for “uncertain/no stricture” (*p* = 0.42). By defining stricture as prestenotic dilation >25 mm, the difference was more considerable for US, Figure 6. Bowel wall thickness measured by US was also associated with increased stiffness: (*ρ* = 0.48, *p* = 0.045). Patients with large ulcers on US had a trend toward a stiffer bowel wall, *E*: 15.9 (13.5–18.7) kPa than those with no ulcers, *E*: 12.1 (9.6–15.3) kPa, *p* = 0.057. Normal (layered) versus disturbed bowel wall stratification on US was not associated with *E*, *p* = 0.62. MRE bowel edema was also not associated with *E* of the stricture (τ_b_ = 0.29, *p* = 0.14). Compound MRE scores were without agreement. No correlation was found between *E* and MaRIA (*ρ* = 0.17, *p* = 0.49). In contrast, there was a moderate association between MEGS and *E* (*ρ* = 0.55, *p* = 0.018, Appendix A).

### 3.3. Stiffness of the Inflammation/Strictures and Elastography, Laboratory Data, and Symptoms

Elastography was only possible in 11 out of the 18 patients. There was no correlation between elastography and biomechanical stiffness, *E*: (*ρ* = −0.05, *p* = 0.87). For predicting strictures, there was a non-significant difference between no strictures 1.75 (1.21–2.29) m/s and “certain” strictures 2.09 (1.23–2.95) m/s. There was a moderate but insignificant association with *E* (r = 0.41, *p* = 0.094) for calprotectin, whereas CRP showed a moderate but significant association (*ρ* = 0.57, *p* = 0.013). Most patients 14/18 (78%) reported severe pain before surgery. No correlation was found between the stricture stiffness and the degree of self-reported pain (τ_b_ = −0.11, *p* = 0.60), HBI (*ρ* = −0.17, *p* = 0.51), or CDAI (*ρ* = −0.14, *p* = 0.58).

## 4. Discussion

We aimed to explore the association between imaging parameters and stiffness of small intestinal inflammation or strictures in CD. Such associations would support the clinical use of specific imaging methods to evaluate CD inflammation and strictures and potentially guide therapy. 

The stricture stiffness was associated with prestenotic dilatation or “certain” strictures on US but not MRE. In contrast, both DCE-MRE initial slope of enhancement and MEGS were moderately and directly associated with stricture stiffness, whereas CEUS was not. This contradicts our hypothesis since we expected an inverse correlation between relative perfusion and stricture stiffness. However, inflammation and fibrosis may co-exist, and the classification of strictures should probably be more detailed than just inflammatory, compound, and fibrotic [25,38]. Fibrosis or even smooth muscle hypertrophy [26] with concomitant severe inflammation and correspondingly high perfusion may explain the direct correlation between perfusion and stiffness. Thus, the term “fibrostenosis” may be imprecise and restrict the correct patient classification [39]. That active inflammation may worsen the bowel stiffness seems likely since stiffness was also associated with increased CRP and fecal calprotectin, although the latter failed to reach statistical significance. Intravenous steroids may reduce the swelling in an acute stricture with bowel obstruction [40,41], further supporting that acute inflammation may influence bowel wall stiffness. Increased bowel wall thickness is directly associated with histologically active disease and fibrosis [31]. 

### Intestinal Biomechanics in Crohn’s Disease

It is well known that clinical symptoms correlate poorly to objective signs of mucosal inflammation in CD [42]. Likewise, our data did not show any correlation between abdominal pain or clinical scores and stricture stiffness. This is anticipated considering the subjective nature of pain and the relatively few patients in our study. However, having a diagnostic tool that can predict a stiff stricture may allow for a better selection of patients undergoing endoscopic balloon dilatation, strictureplasty, or resection. This is important since cross-sectional imaging modalities cannot always differentiate between inflammation and fibrosis [11,26,43]. 

Stidham et al. utilized strain elastography to predict bowel wall elasticity in a rat model of colitis and later in seven CD patients undergoing surgery for intestinal strictures [44]. The elasticity reference standard was tissue stiffness expressed as Young’s modulus measured by a micro-elastometer [45,46]. A clear difference was found between normal and fibrotic tissue. Fraquelli et al. [47] also investigated CD strictures with elastography. They used a quantitative tissue analysis of fibrotic content as a gold standard rather than biomechanical properties. Nevertheless, they found a direct association between ultrasound Elastography and fibrotic content in the strictures. Havre et al. evaluated strain elastography on human intestinal specimens and found no association with increasing histological inflammation. The authors also concluded that reproducibility was moderate, and strain elastography did not differentiate between CD and adenocarcinoma [48]. Baumgart et al. compared strain elastography with tissue tensiometry by applying a 250 g weight and registering the stretch percentage. Unfortunately, the authors did not specify when the tissue tensiometry was performed. In a recent study comparing Shear Wave Elastography with histopathology, Lu et al. found only a moderate correlation between elasticity and muscular hypertrophy [26]. 

All the studies mentioned above either investigated tissue elastography or longitudinal stress-strain relationship for a section of the intestine. The more important circumferential wall stress was not studied. We found no statistically significant associations between shear wave elastography and circumferential wall stress or presence vs. absence of a stricture. These negative findings could result from a small sample size with high standard deviations. However, numerous issues with shear wave elastography assessment in the intestines may hamper any meaningful association with true stiffness or even histological composition. First, the success rate of the modality was low, with a high SD of the ten measurements. Secondly, fitting the ROI inside the bowel wall may be difficult when the bowel is only moderately thickened. In addition, reproducibility is affected by the depth of measurement and the absolute stiffness of the examined element [49,50]. The number of outliers can be between 50–80% [49] as in vivo measurements of organs are more complex to evaluate compared to phantoms [50]. In contrast to methods used in vivo to obtain tissue stiffness measures, such as elastography, the classical biomechanical methods we used ex vivo are more accurate and reproducible [12,13,14,15]. 

Strictures, defined by US, had a significantly higher *E* than luminal inflammation or uncertain strictures, and this was not the case for MRE. Although oral contrast consumption before an US scan may be superior to non-contrast intake [51], we believe that any prestenotic dilatation in the fasting patient is a clear sign of a stricture. We adopted the proposed definition of stricture by Bettenworth et al. [11] and the latest EFSUMB recommendations for CD [18] using >25 mm as the cut-off for prestenotic dilatation as a post-hoc analysis. By applying this definition, the difference in stiffness became more pronounced. The amount of prestenotic dilatation in enterography can be more challenging to determine since the bowel is already distended. Both bowel wall edema and contrast enhancement pattern on DCE-MRE, both parameters that correlate with active inflammation, were associated with stiffness. In addition to chronic changes such as fibrosis and muscular hypertrophy, active inflammation likely causes increased stiffness [43,52]. 

Our study has limitations. The distension was not performed in vivo and without the standard hormonal and neurogenic feedback mechanisms. Thus, we studied passive mechanical properties, which is advantageous since the structure is made of passive elements such as collagen. Active contraction of smooth muscle cells within a stricture will increase wall stiffness. Our study could theoretically have been conducted intra-operatively, and however, this would have increased surgery duration and the risk of contamination during the procedure. In vitro and ex vivo studies can use classical biomechanical methods, including imaging with US having high accuracy [12,13,14,15].

In the analysis of both DCE-MRE and CEUS, we did not take the arterial input function into account [53]. This may limit the inter-patient correlation [31]. Jirik et al. have proposed a way to overcome this, but reproducibility and consensus for applying their method have not been established in CD [54]. Although prestenotic dilatation and bowel wall pattern are objective findings, we applied several semiquantitative assessments for which reproducibility was not tested. Further, we did not apply a power calculation as this was a proof-of-concept study. The sample size was relatively small and based on pragmatic considerations.

## 5. Conclusions

We reported data from in vivo imaging and studied their association with the circumferential stiffness of bowel inflammation and strictures in CD. Surprisingly, direct correlations were found between the stricture stiffness and several indicators of active inflammation, such as the initial slope of enhancement on MRE, the MEGS, and CRP. Stiffness also correlated well with prestenotic dilatation on US without oral contrast agents, potentially a new biomarker.

## Figures and Tables

**Figure 1 diagnostics-12-01370-f001:**
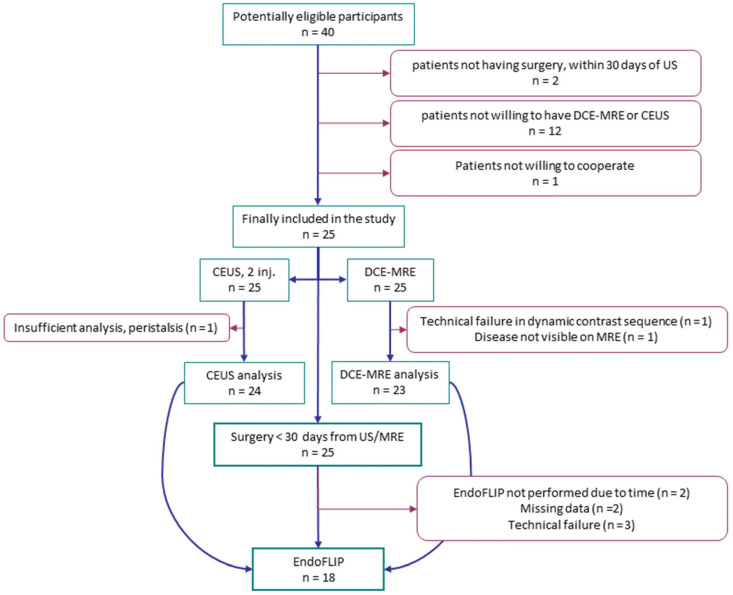
Flow chart of patients included in the study.

**Figure 2 diagnostics-12-01370-f002:**
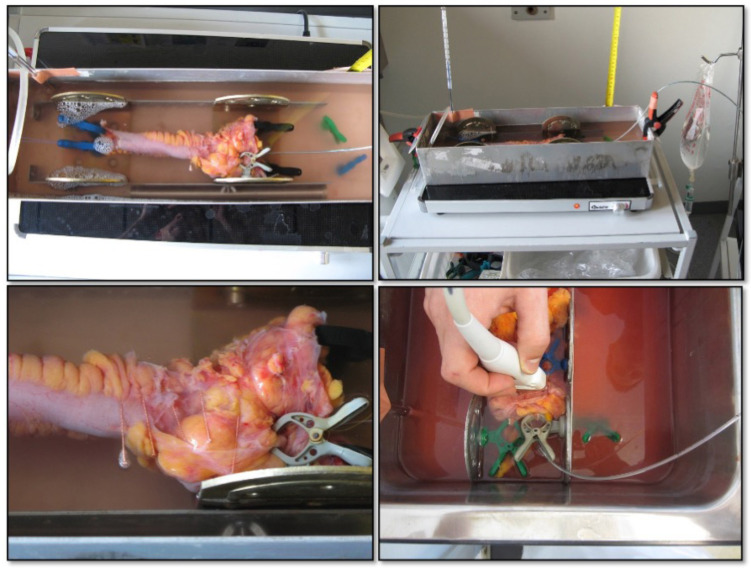
Experimental setup. Water container with oxygenated Krebs solution and the specimen stretched to original length using magnetic clamps (**top left**). The water container was placed on a table heater (37 °C) with a ruler and bag for applying pressure (**top right**). Acupuncture needles inserted as location markers for ultrasound (**bottom left**). Ultrasound probe during scanning the bowel segment (**bottom right**).

**Figure 3 diagnostics-12-01370-f003:**
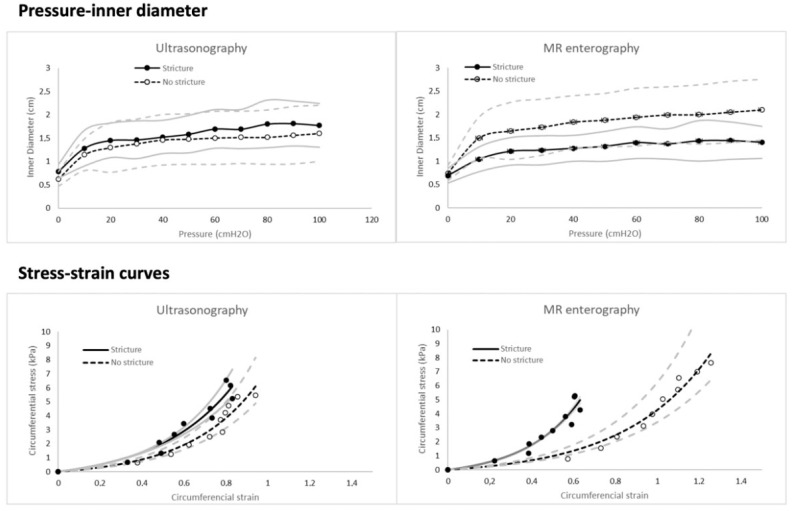
The averaged inner diameter-pressure curves (**top**) and averaged circumferential stress-strain curves (**bottom**) for strictures preoperatively assessed as “stricture” (solid lines) or “non-stricture” (dashed lines) with ultrasonography (**left**) and MR enterography (**right**). Dots are averaged data, and solid lines are curve-fitted. Grey lines are confidence intervals.

**Figure 4 diagnostics-12-01370-f004:**
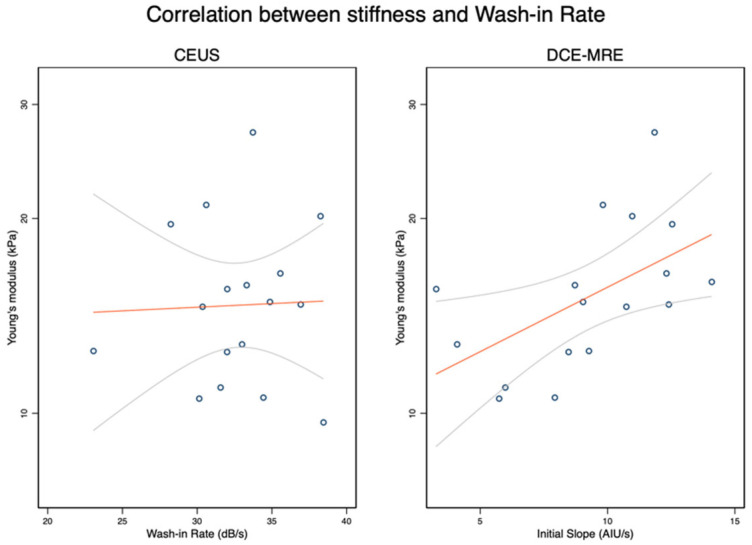
The stiffness of small intestinal strictures is associated with the initial slope of Dynamic Contrast-Enhanced MR Enterography (DCE-MRE) (**right**), spearman’s *ρ* = 0.63, *p* = 0.007, but not with wash-in rate of Contrast-Enhanced Ultrasonography (CEUS) (**left**), spearman’s *ρ* = 0.08, *p* = 0.76.

**Figure 5 diagnostics-12-01370-f005:**
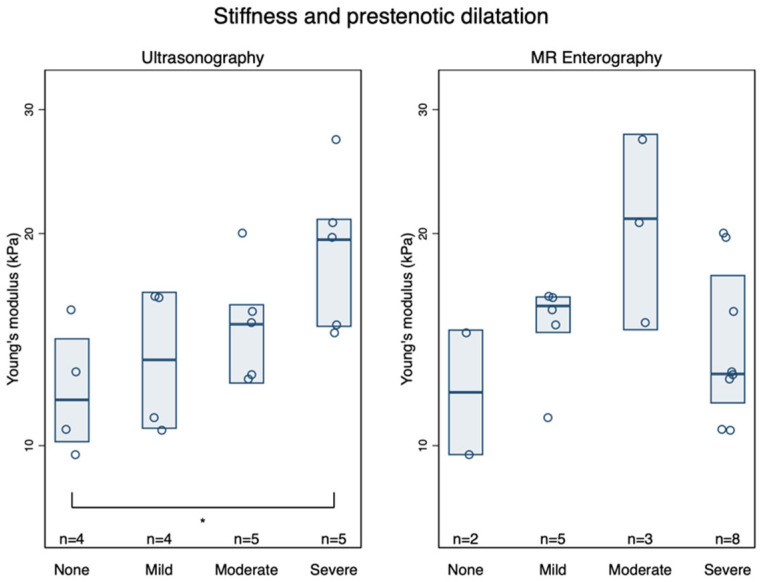
Associations between stiffness of small intestinal strictures and the degree of prestenotic dilatation on ultrasonography (τ_b_ = 0.43, *p* = 0.02) (**left**) and MR Enterography (τ_b_ = 0.01, *p* = 1.0) (**right**). Boxes are median and inter-quartile ranges. * *p* < 0.05.

**Figure 6 diagnostics-12-01370-f006:**
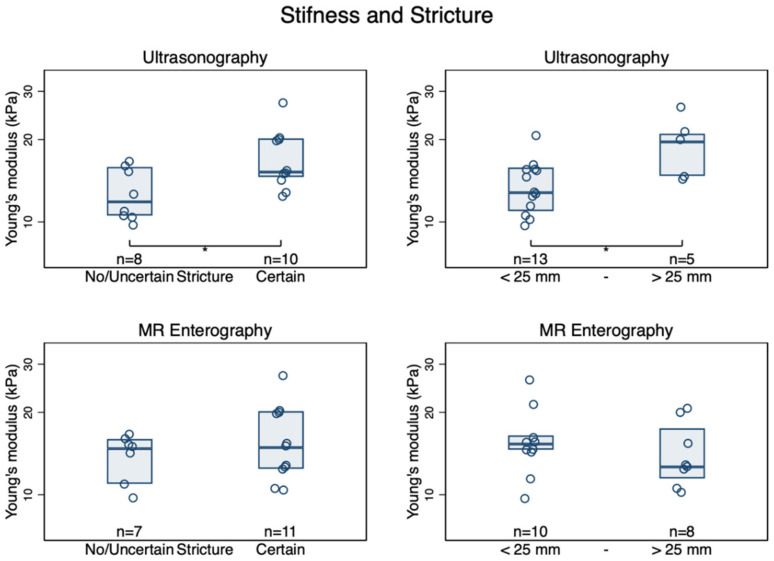
Box plot of circumferential stiffness in small intestinal Crohn’s disease strictures defined as certain vs. no/uncertain by ultrasonography (**top left**) and MR enterography (**bottom left**). When applying definitions of prestenotic dilatation of >25 mm for becoming a certain stricture on ultrasonography (**top right**) and MR Enterography (**bottom right**), Boxes are median and inter-quartile ranges. * *p* < 0.05.

## Data Availability

Data available on request due to restrictions, e.g., privacy or ethical. The data presented in this study are available on request from the corresponding author.

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
