# Peer review of "Biomechanical Properties of Strictures in Crohn’s Disease: Can Dynamic Contrast-Enhanced Ultrasonography and Magnetic Resonance Enterography Predict Stiffness?"

_diagnostics, 2022, doi:10.3390/diagnostics12061370_

Round 1

Reviewer 1 Report

Thank you for this interesting paper.

Crohn's strictures are difficult to manage and your work helps increase our understanding. 

The topic is original and it does address a specific gap in the field. It adds more detail on how to differentiate types of strictures which is important clinically.
I think the methodology is well planned and limitations are discussed.
Conclusions are consistent with the evidence but abstract needs to be reworded. The references are appropriate as well and the diagrams and photo are very helpful too.

Some comments

Abstract needs to be edited  and should reflect what you write in your conclusion ie. stiffness correlating with inflammation 

Please add small numbers into limitations of the study.

Minor editing eg leaving a space between text and reference needed please. 

Author Response

We thank Reviewer 1 for reviewing the manuscript and providing good feedback.

1) Abstract needs to be edited  and should reflect what you write in your conclusion ie. stiffness correlating with inflammation 

Thank you for adding this comment. We have now highlighted that the initial slope of enhancement and the MRE Global Score are parameters of active disease.

Please add small numbers into limitations of the study.

Thank you for highlighting this apparent issue. We have now added the following text to the limitation section:

 Further, we did not apply a power calculation as this was a proof-of-concept study. The sample size was relatively small and based on pragmatic considerations.

Minor editing eg leaving a space between text and reference needed please. 

Thank you very much for this comment. The manuscript looks better this way.

We edited the English language throughout for clarification and readability.

Reviewer 2 Report

The paper: ” Biomechanical properties of strictures in Crohn’s Disease: Can dynamic contrast-enhanced ultrasonography and magnetic resonance enterography predict stiffness?” presents novel approach on evaluating stricture stiffness in CD. The problem is very important- we still do not have antifibrotic drugs.

The study was well designed and the proper techniques were used: CEUS, DCE-MRE, SWE. The surgical specimens were thoroughly checked for stiffness.  The results showed notable role of DCE-MRE in contrast to insufficient performance by SWE.

Paragraph 2.2 is in bold font- please correct.

I can recommend the paper for publication.

Author Response

We are grateful for the comments by reviewer 2. 

Paragraph 2.2 is in bold font- please correct.

Thank you for highlighting this. It is now corrected.

I can recommend the paper for publication.

Thank you very much.

We edited the English language throughout for clarification and readability.